# Preparation of Degradable Superhydrophobic Mg/P/Z/F/H Composite Materials and Their Anticorrosion

Zhongxian Xi [1,2], Chengqing Yuan [1,2,*], Xiuqin Bai [1,2], Chun Wang [3] and Anne Neville [3]

1   Reliability Engineering Institute, National Engineering Research Center for Water Transport Safety, MOST, Wuhan 430063, China; xizhongxian@126.com (Z.X.); xqbai@whut.edu.cn (X.B.)
2   School of Energy and Power Engineering, Wuhan University of Technology, Wuhan 430063, China
3   Institute of Functional Surfaces, Department of Mechanical Engineering, University of Leeds, Leeds LS2 9JT, UK; C.Wang@leeds.ac.uk (C.W.); A.Neville@leeds.ac.uk (A.N.)
*   Correspondence: ycq@whut.edu.cn; Tel.: +86-027-86582035

**Abstract:** In this study, the degradable superhydrophobic Mg/P/Z/F/H (magnesium/poly(-caprolactone)/zinc oxide/1H,1H,2H,2H-perfluorodecyltriethoxysilane (PFDTES)/heating process) composite materials were prepared through dip-coating method and heating process, for enhancing the corrosion resistance of the AZ91D magnesium alloys. The electrochemical measurements revealed that the Mg/P/Z/F/H materials significantly improved the corrosion resistance of the magnesium alloys in 3.5 wt.% NaCl. The Mg/P/Z/F/H composite materials exhibited efficient self-cleaning properties, good adhesion strength, and stability in wet atmosphere.

**Keywords:** superhydrophobic surface; heating process; dip-coating; poly($\varepsilon$-caprolactone) (PCL); self-cleaning

## 1. Introduction

With special solid–liquid adhesion, self-cleaning [1–11], anti-icing [4,6–11], anti-corrosion [5,9,12,13], antibacterial, and other outstanding characteristics [14–16], the multifunctional super-antiwetting surfaces have received extensive research attention with respect to preparations and applications. The research on the superhydrophobic materials focuses on the undegradable polymers (epoxy resin, polyurethane, polytetrafluoroethylene, polyvinyl chloride, perfluoropolyalkylether, etc.) and various metal alloys (Fe, Cu, Ni, Co) containing heavy metal ions [17–22]. With the widespread application of metals and undegradable polymers, the environmental risks of various heavy metal (especially Pb, Cu, Cd, Cr and Ni) and the plastic particle pollution are becoming serious under different corrosive medium. In this study, the environmentally degradable materials (magnesium, polycaprolactone, and ZnO) were used to prepare the multifunctional superhydrophobic materials, so as to overcome the limitations of the existing materials.

As one of the lightest metals, with a density two-thirds that of aluminum and one-quarter that of steel, magnesium alloys exhibit a significant potential to improve the system performance and energy efficiency in aerospace, automotive, shipbuilding, mobile electronics, and bioengineering applications owing to the excellent chemical, mechanical, biological, and physical properties of magnesium [23–27]. However, because of magnesium's poor corrosion resistance especially in the corrosive medium environment of Cl⁻, magnesium alloys' applications are limited [27–31]. Furthermore, containing rare heavy metal elements such as Cu, Ni, and Pb, magnesium alloys are environmentally friendly and used as biodegradable metals [25]. As a typical aliphatic polyester, poly($\varepsilon$-caprolactone) (PCL) is nontoxic and ecofriendly to living organisms, and is widely used in food packaging and the pharmaceutical industry [25,32–36]. ZnO powder is used in electronic and optoelectronic devices, solar cells, and pharmaceutical applications due to their high safety, low price, extraordinary opto-electronic characteristics, and non-polluting character [35,37,38].

Widespread applications of degradable materials can decrease environmental pollution associated with abandoned ships in sea, scrapped cars in wild areas, excess building materials, etc. Compared with plasma electrolytic oxidation (PEO) [39,40], hydrothermal treatment [41], and electrodeposition [42] techniques, non-degradable polymer coatings on the magnesium alloys, such as polypropylene [43], polyvinyl chloride [44], and epoxy resin [45], cannot degrade and may bring potential harmful effects on organisms.

In this study, degradable and superhydrophobic M/P/Z/F/H composite materials were prepared through the dip-coating method and heating process to solve the poor corrosion resistance of magnesium. As a widely applied and simple surface treatment technology, the dip-coating process can produce relatively adherent, stable, and uniform films on the material surfaces [46–49]. The rough Mg/P/Z/F/H structure was constructed by using the dip-coating method to effectively protect the magnesium alloy. Furthermore, the heating process for 30 min at 50 °C was applied to repair the defects on the surface of the composite Mg/P/Z/F materials as well as to rearrange the PCL and PFDTES molecules to transform the composite surface from hydrophobic (96.5°) to superhydrophobic (159.0°). The surface morphology, microstructure, adhesion strength, corrosion resistance, and self-cleaning of the superhydrophobic Mg/P/Z/F/H composite materials were subsequently characterized.

## 2. Experimental Section

### 2.1. Materials

The following chemicals were used without further modification: ethanol (Tianjin Damao Chemical Reagent Factory, Tianjin, China), dichloromethane (DCM) (Qingmei Chemical Reagent Factory, Shenzhen, China), 1H,1H,2H,2H-perfluorodecyltriethoxysilane (PFDTES) (Guangzhou Hongcheng Biotechnology Science and Technology Co. Ltd., Guangzhou, China, 97%), PCL (Mw: 80,000, Solvay Specialty Polymers, Qingdao, China), and ZnO powders (30 nm, Beijing Shenghehaoyuan Tech. Co. Ltd., Beijing, China). The die-cast AZ91D magnesium alloy (Dongguan Qingmei Metal Materials Co. Ltd., Dongguan, China) was used as the substrate material.

### 2.2. Sample Preparation

In this study, the die-cast Mg alloy AZ91D with the chemical composition (wt.%) of 8.77% Al, 0.74% Zn, 0.18% Mn, and 90.31% Mg was cut into pieces of dimensions 25 mm × 25 mm × 5 mm. Prior to use, the samples were polished by the silicon carbide papers from 150 to 1000 mesh. Subsequently, the ultrasonic cleaning in acetone and de-ionized water rinsing were performed. Finally, the samples were dried at 60 °C.

### 2.3. Preparation of Mg/P/Z/F/H Composite Materials

By dip-coating method, the Mg/P/Z/F composite materials were prepared with suitable concentrations of PCL solutions as was shown in Scheme 1.

The PCL (5 wt.%) granules were dissolved in 60 mL dichloromethane (DCM) under magnetic stirring for 5 h. Subsequently, the ZnO powder was added to the PCL solution (5 wt.%) and stirred continuously. It was followed by the addition of 1.5 mL PFDTES and continuous stirring for 10 h. The prepared samples were immersed in the mixed solutions for 30 s and pulled out of the solution at a speed of 2 mm/s, leading to the formation of the Mg/P/Z/F composite materials. Finally, the Mg/P/Z/F/H materials were generated after heating the Mg/P/Z/F materials at 50 °C for 30 min. Table 1 presents the details of the different layers (Mg/P, Mg/P/Z, Mg/P/Z/F, and Mg/P/Z/F/H).

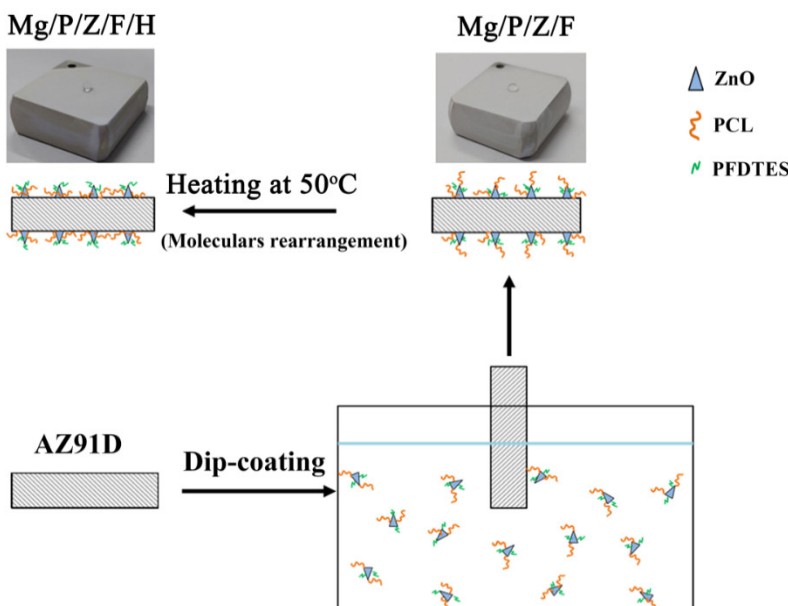

**Scheme 1.** The schematic illustration of the fabrication of the superhydrophobic Mg/P/Z/F/H composite materials.

**Table 1.** The coatings developed using the different groups of samples.

| Substrates | Dip-Coating Concentrations | Treatment Condition |
|---|---|---|
| Mg/P | 5 wt.% PCL | None |
| Mg/P/Z | 5 wt.% PCL; 5 wt.% ZnO | None |
| Mg/P/Z/F | 5 wt.% PCL; 5 wt.% ZnO; 1.5 mL PFDTES | None |
| Mg/P/Z/F/H | 5 wt.% PCL; 5 wt.% ZnO; 1.5 mL PFDTES | 50 °C, 30 min |

### 2.4. Surface Characterization and Property Tests

The surface characterization was performed via scanning electron microscopy and energy dispersive spectrometry at 10 kV (VEGA3, Tescan China Ltd., Shanghai, China). The water contact angles were measured at ambient temperature using an optical contact angle meter (Dataphysics OCA 15EC, Filderstadt, Germany) by placing a 5 μL water drop on the surface. The Fourier transform infrared (FTIR, NEXUSFT-870, Thermo Fisher Scientific, Waltham, MA, USA) spectra were recorded in the range 400–4000 cm$^{-1}$. The electrochemical tests were performed by using an electrochemical workstation (CorrTest CS350, Wuhan Corrtest Instruments Corp., Ltd., Wuhan, China).

## 3. Results and Discussion

### 3.1. Wetting Behaviors

The wetting behaviors of Mg/P (with PCL), Mg/P/Z (with PCL and ZnO), Mg/P/Z/F (with PCL, ZnO and PFDTES), and Mg/P/Z/F/H (with PCL, ZnO and PFDTES) were characterized in this study. The Mg/P, Mg/P/Z, Mg/P/Z/F, and Mg/P/Z/F/H material surfaces exhibited varying extent of hydrophobicity (Figure 1). In Figure 1a, the ecofriendly and degradable PCL coating was noted to be hydrophilic, and the CA value of the Mg/P surface was 65.5 ± 1.1° (Figure 1a). With the addition of ZnO powder, the micro-nano rough structure could be constructed, and the CA value of the Mg/P/Z material increased from 65.5 ± 1.1° to 98.6 ± 1.5° (Figure 1b). Subsequently, the Mg/P/Z/F materials were prepared by mixing PFDTES at room temperature. However, the CA value of the Mg/P/Z/F surface was similar as that of Mg/P/Z. The room temperature mixing of PFDTES did not impact the wetting behavior evidently and the CA value of Mg/P/Z/F was noted to be 96.5 ± 1.8° (Figure 1c). After 30 minutes of the heating process at 50 °C of

the Mg/P/Z/F, the Mg/P/Z/F/H surface was obtained, with CA value of 159.0 ± 1.6°, as shown in Figure 1d.

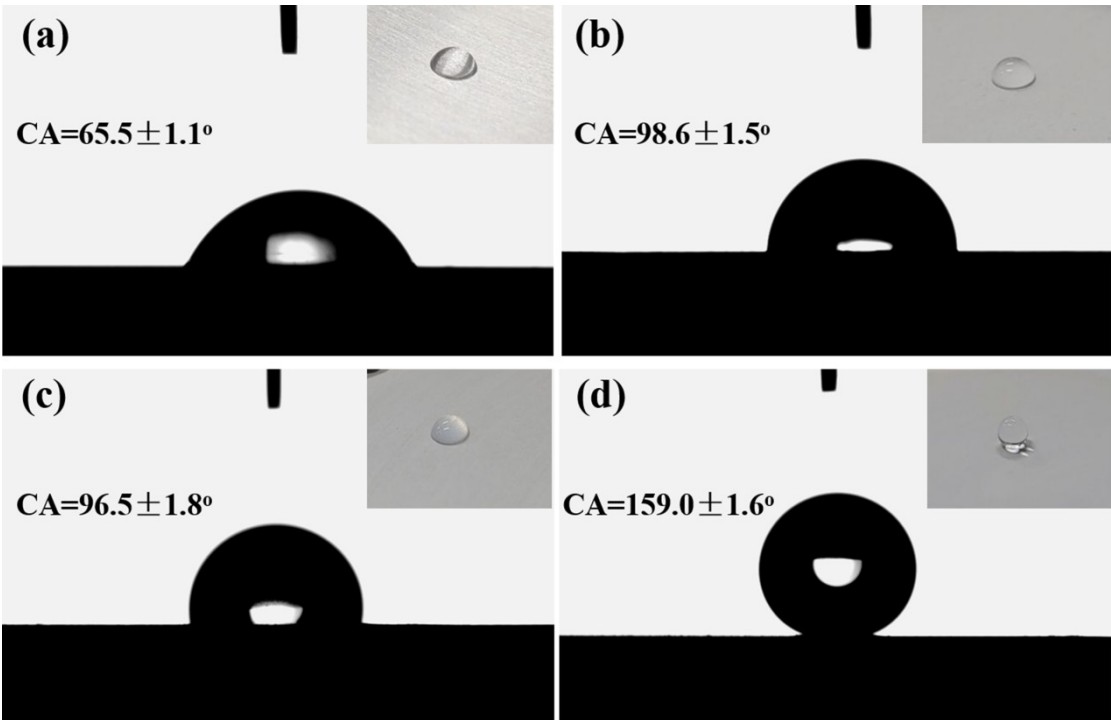

**Figure 1.** The images of the water droplet contact angles on Mg/P (**a**), Mg/P/Z (**b**), Mg/P/Z/F (**c**), and Mg/P/Z/F/H (**d**) samples.

### 3.2. Surface Characteristics

The SEM and EDS analyses of Mg/P/Z, Mg/P/Z/F, and Mg/P/Z/F/H are presented in Figure 2. Mg/P exhibited high porosity, thus, leading to an ineffective protection of the magnesium alloy substrates against the corrosive medium [25]. Mg/P/Z exhibited a heterogeneous surface with micro-scaled roughness and high porosity (Figure 2a,d). As shown in Figure 2b,e, the Mg/P/Z/F material possessed a low porosity, and the ZnO particles were evenly bonded together by PCL. However, numerous gully defects were observed on the surface of the Mg/P/Z/F composite materials. After heating process for 30 min, at 50 °C, the gully defects were observed to be repaired, and the Mg/P/Z/F/H surface became superhydrophobic (Figure 2c,f). In a former study, after heating Mg/P/Z/F at 35 and 45 °C for 30 min, the CA value of the composite surface did not change significantly and remained around 98°. In addition, the EDS analysis of Mg/P/Z (Figure 2g), Mg/P/Z/F (Figure 2h), and Mg/P/Z/F/H (Figure 2i) illustrated that without the heating process, the PFDTES content on the surface of Mg/P/Z/F was low (Si: 0.28 wt.%; F: 2.14 wt.%). However, after heating at 50 °C, the content of PFDTES on the surface of Mg/P/Z/F/H was observed to increase (Si: 0.34 wt.%; F: 4.26 wt.%).

The composition of Mg/P, Mg/P/Z, Mg/P/Z/F, and Mg/P/Z/F/H was further analyzed via Fourier transform infrared spectroscopy (FTIR), as shown in Figure 3. The vibrations at approximately 2956 and 2875 cm$^{-1}$ were attributed to CH$_3$ [$v$ (CH$_3$)] in Mg/P, Mg/P/Z, Mg/P/Z/F, and Mg/P/Z/F/H. Further, the CF$_3$ stretching vibrations [$v$ (CF$_3$)] appeared at 1172 cm$^{-1}$ in Mg/P/Z/F and Mg/P/Z/F/H [50,51]. The Zn-O stretching vibration at 832 cm$^{-1}$ in Mg/P/Z/F/H could explain that the PCL transfer occurred on ZnO after the heating process [52]. Moreover, the FTIR tests were consistent with the EDS results and after heating process the ZnO powders were exposed.

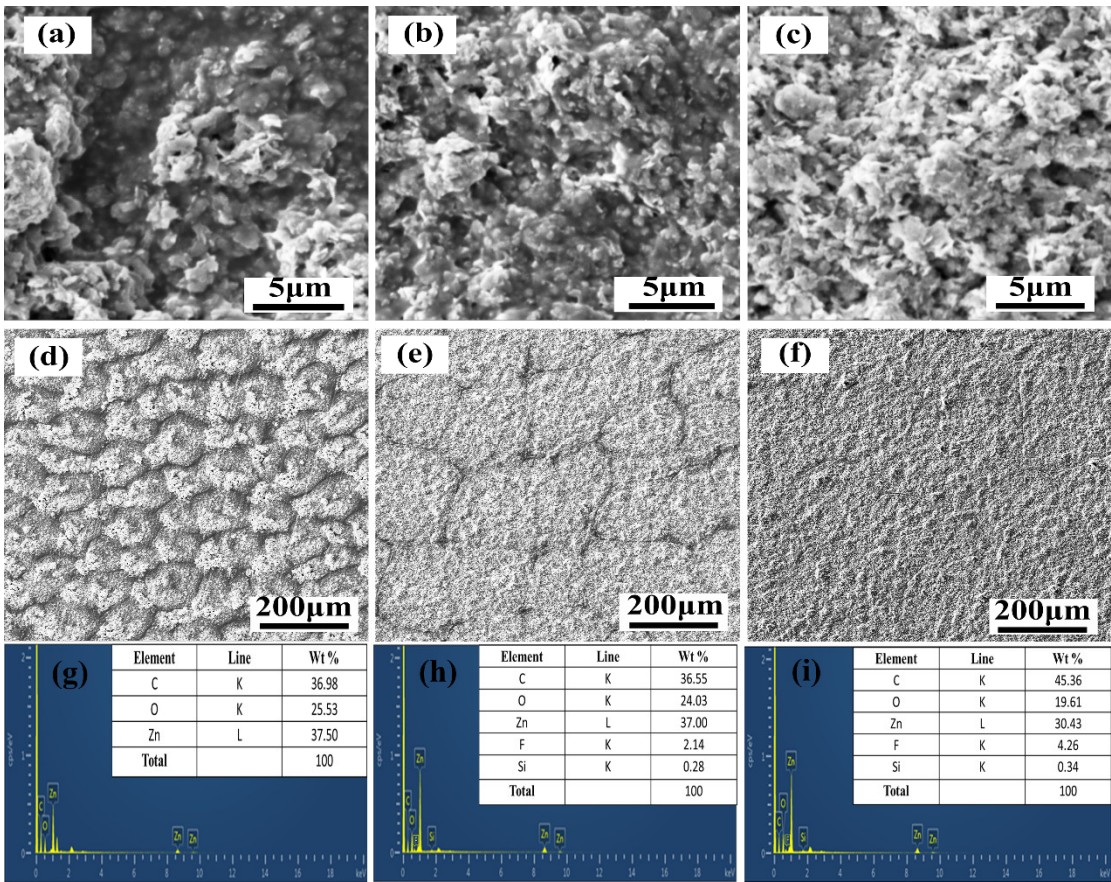

**Figure 2.** The SEM micrographs of Mg/P/Z (**a**,**d**), Mg/P/Z/F (**b**,**e**), and Mg/P/Z/F/H (**c**,**f**); and the EDS analysis of Mg/P/Z (**g**), Mg/P/Z/F (**h**), and Mg/P/Z/F/H (**i**).

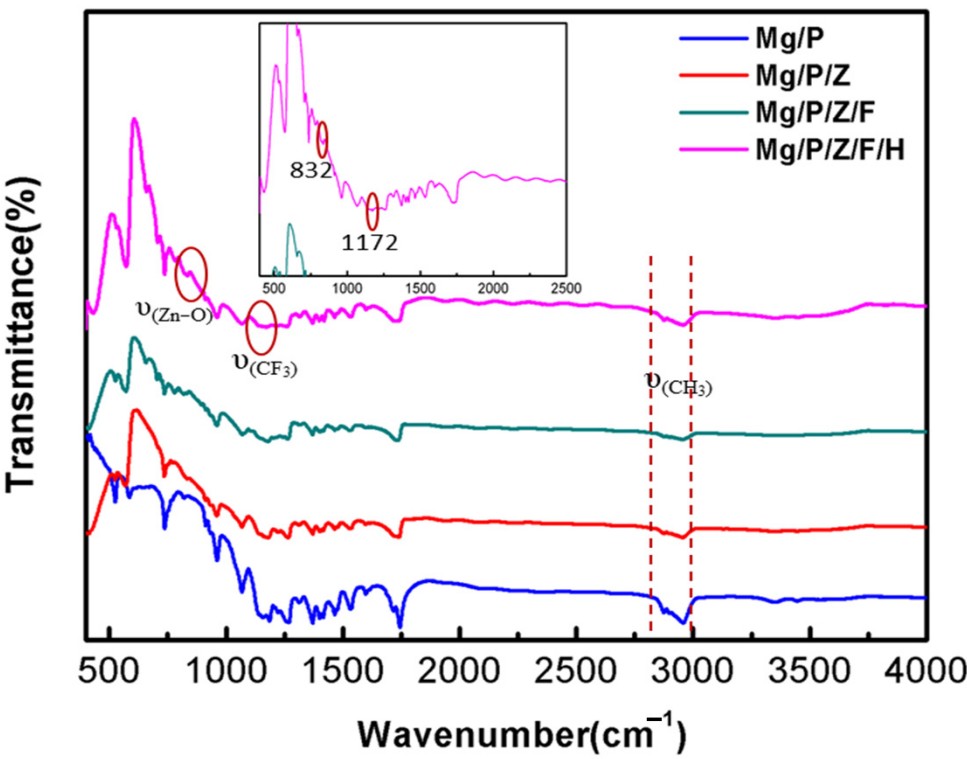

**Figure 3.** The FTIR analysis of Mg/P, Mg/P/Z, Mg/P/Z/F, and Mg/P/Z/F/H composite materials.

The molecules rearrangement mechanism was proposed to explain the hydrophobic and superhydrophobic change process at the softening temperature of PCL (50 °C), as shown in Figure 4. At the softening temperature (50 °C, about 9 °C lower than the melting point of PCL (Mw: 80,000)), the PCL molecules could move more violently and freely. At the same time, the incarceration ability of PCL was weakened. After 30 minutes of the heating process, the PFDTES and PCL molecules underwent molecules rearrangement, and the PFDTES molecules were exposed on the surface of the composite Mg/P/Z/F/H materials to form the superhydrophobic surface (Figure 4). Furthermore, the frequent molecular motion could repair the gully defects of the Mg/P/Z/F/H materials' surface (Figure 2c,f).

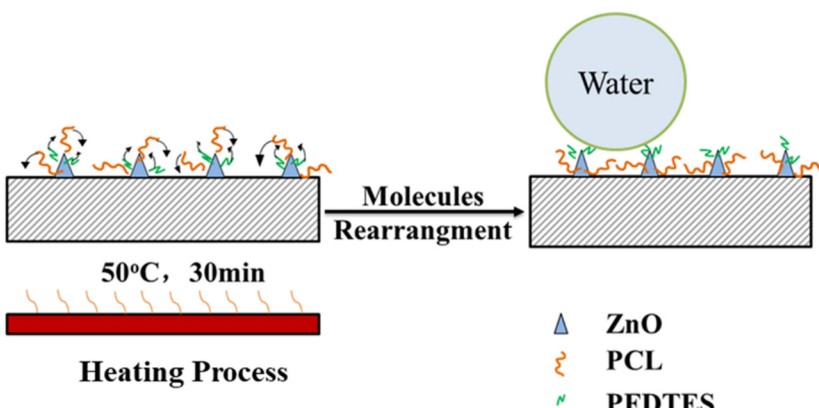

**Figure 4.** The process of Mg/P/Z/F transforming to Mg/P/Z/F/H at 50 °C.

### 3.3. Electrochemical Corrosion Behavior

The electrochemical impedance spectra (EIS) were acquired using a three-electrode system in 300 mL 3.5 wt.% NaCl. A sample surface area of 1 cm$^2$ was exposed. A saturated calomel and a platinum mesh electrode were used as the reference and counter electrodes, respectively. In Figure 5, the EIS plots of the five groups (Mg, Mg/P, Mg/P/Z, Mg/P/Z/F, and Mg/P/Z/F/H) were reflected in 3.5 wt.% NaCl after a proper stabilization time (about 40 to 60 min). In the Nyquist plane, the impedance value of the working electrode is represented by the diameter of the capacitive loop [25]. As shown in Figure 5, the Mg/P/Z/F/H sample demonstrated the best anticorrosion performance. The value of Rct for Mg/P/Z/F/H (>1.5 × 10$^7$ Ω·cm$^2$) was about 2 × 10$^4$ times larger than that of the Mg sample (750 Ω·cm$^2$). The EIS analysis of Mg/P/Z/F/H indicated that the superhydrophobic surface after the heating process led to a desirable anti-corrosion performance.

Besides, the corrosion potential (E$_{corr}$), corrosion current density (i$_{corr}$), and corrosion rate (CR) were calculated by using the Tafel extrapolation method (Figure 6 and Table 2). The polarization curves could be a typical indication of the materials' stability. The substrate corrosion current density of the composite materials was observed to decrease significantly with the repair of the gully defects and molecular rearrangement after the heating process. Based on the Cassie-Baxter state, the air trapped on the micro/nanostructured surface improved the hydrophobicity and corrosion resistance of Mg/P/Z/F/H [47,48]. Mg/P/Z/F/H exhibited the best corrosion resistance, with the corrosion rate determined to be 1.9152 × 10$^{-3}$ mm/y. The addition of ZnO obviously decreased the corrosion resistance of Mg/P/Z as the ZnO powder disturbed the continuity of the PCL layer (Figure 2d). In addition, the discontinuity of the layer was modified after mixing PFDTES (Figure 2e). Furthermore, the main components of Mg/P/Z/F/H are PCL, ZnO, and magnesium alloys with the degradation and biocompatibility [53–56]. In addition, after 6 h immersion in 3.5 wt.% NaCl, I$_{corr}$ of Mg/P/Z/F/H was about 1.1703 × 10$^{-7}$ A/cm$^2$, whereas the E$_{corr}$ value was observed to be −1.1866 V. This indicated that Mg/P/Z/F/H was stable in the NaCl corrosive medium.

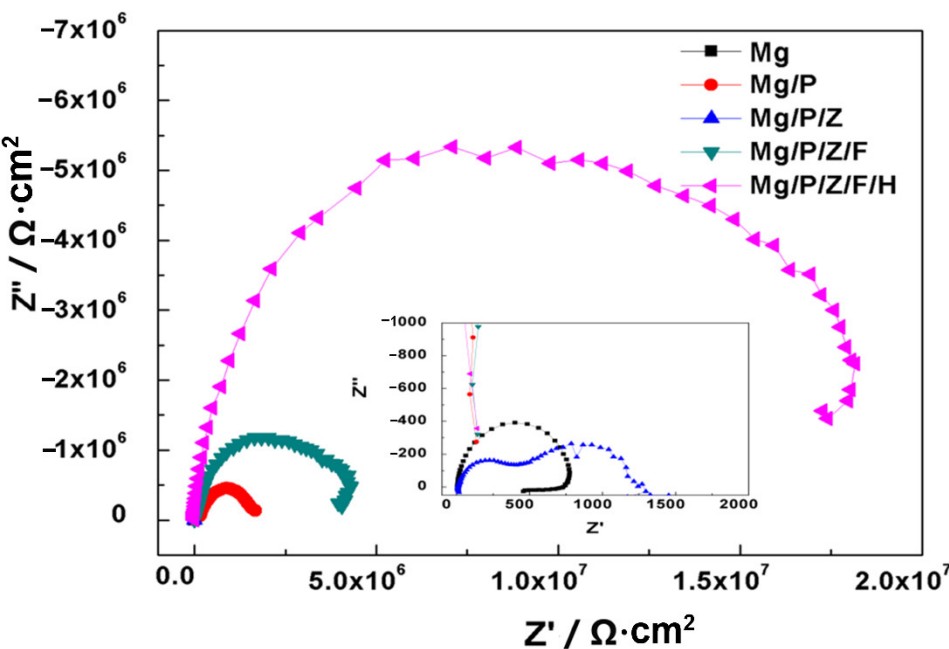

**Figure 5.** The electrochemical impedance spectroscopy (EIS) of the samples in 3.5 wt.% NaCl.

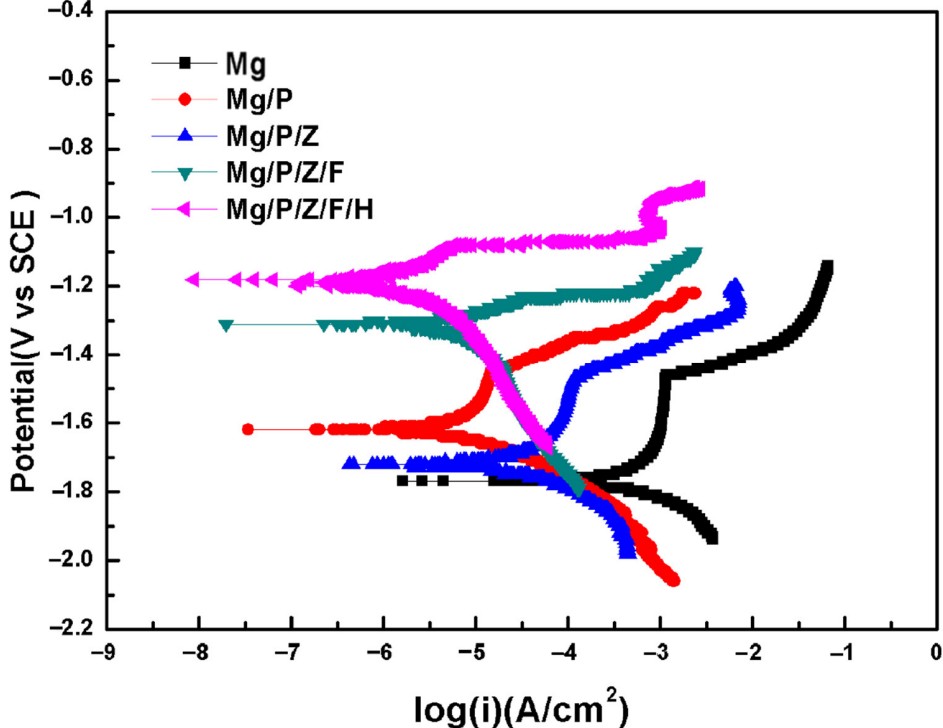

**Figure 6.** The polarization curves of Mg, Mg/P, Mg/P/Z, Mg/P/Z/F, and superhydrophobic Mg/P/Z/F/H.

**Table 2.** The electrochemical parameters of samples.

| Substrates | $I_{corr}$ (A/cm$^2$) | $E_{corr}$ (V) | Corrosion Rate (mm/y) |
|---|---|---|---|
| Mg | $1.0237 \times 10^{-3}$ | $-1.7694$ | 22.0580 |
| Mg/P | $1.7464 \times 10^{-5}$ | $-1.6203$ | 0.3763 |
| Mg/P/Z | $1.6043 \times 10^{-5}$ | $-1.7200$ | 0.3457 |
| Mg/P/Z/F | $7.1288 \times 10^{-6}$ | $-1.3073$ | 0.1536 |
| Mg/P/Z/F/H | $8.7914 \times 10^{-8}$ | $-1.1752$ | $1.9152 \times 10^{-3}$ |

*3.4. Interface Model and Anticorrosion Mechanism*

The water drops can have two states on a rough surface: Wenzel state and Cassie-Baxter state [57,58]. Here, owing to a low sliding angle, the superhydrophobic surface could be explained by the Cassie-Baxter state. The micro/nanostructured surface enhanced the hydrophobicity owing to the trapped air. Further, heating process at 50 °C could make the PFDTES molecules exposed on the surface of Mg/P/Z/F/H to form a continuous air film, as shown in Figure 7. The continuous air film separated the substrate from the corrosive medium, thus, improving the surface hydrophobicity and corrosion resistance of Mg/P/Z/F/H.

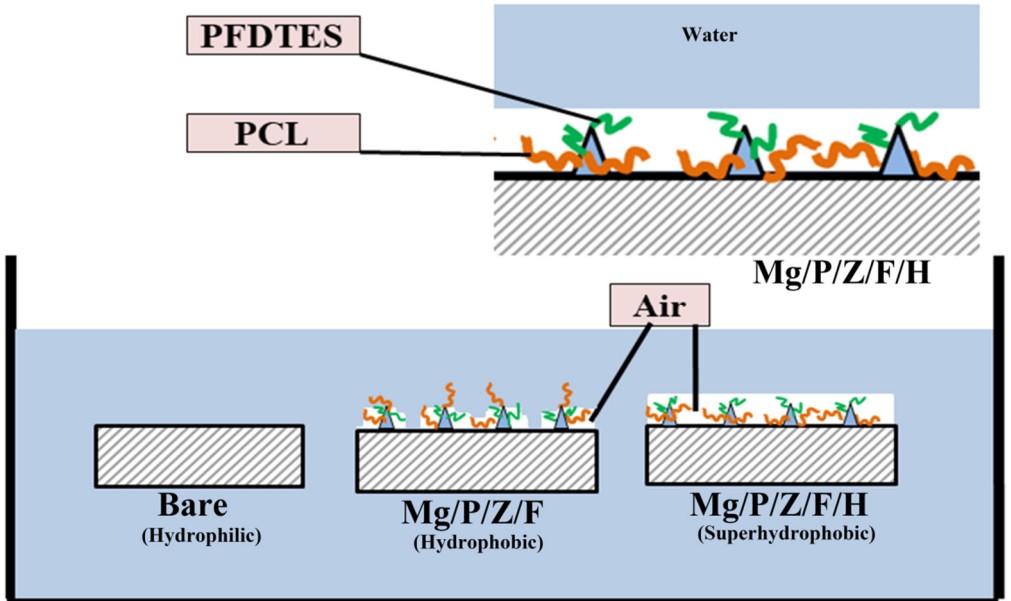

**Figure 7.** The anticorrosion mechanism of the superhydrophobic surface in the corrosive medium.

*3.5. Stability and Adhesion*

For the practical applications, the stability of the materials is an important parameter. For this purpose, the mechanical and superhydrophobic stability of the material surface needs to be considered. The stability of the superhydrophobic surface was tested in a wet atmosphere at room temperature, as shown in Figure 8a. The superhydrophobic Mg/P/Z/F/H surface was noted to be stable, and the CA value was over 155° after 168 h in the wet atmosphere (Figure 8b, the variation of CA was between +1.8° and −1.6°).

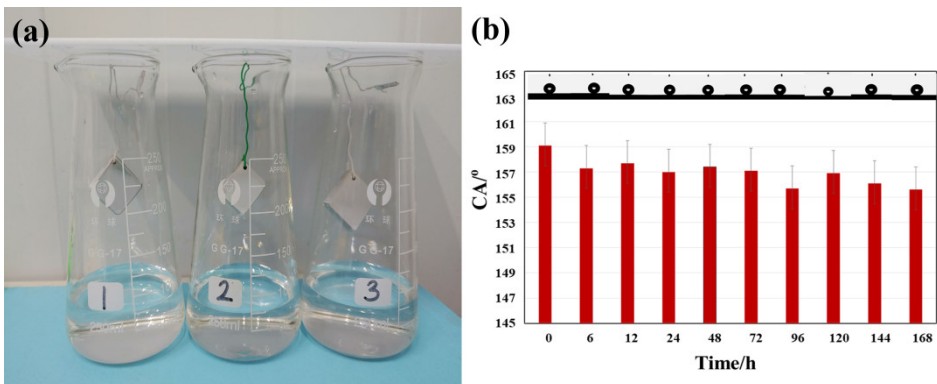

**Figure 8.** The stability test in the wet atmosphere (**a**) and the corresponding stability data (**b**).

The mechanical adhesion stability of the surface was evaluated by using the Scribe-Grid Test (ASTM D 3359-78) [25]. Figure 9a,b present the optical images of the superhydrophobic Mg/P/Z/F/H surface before (Figure 9a) and after (Figure 9b) the tape test. No detachment or delamination of the film was observed at the edges and within the square lattice, as shown in Figure 9a,b. The adhesion of the composite film corresponded to the 4B classification according to ASTM D 3359-78. These findings confirmed that the strong adhesion of the superhydrophobic Mg/P/Z/F/H for effective application in the engineering and industrial conditions. The efficient adhesion can effectively protect the substrate from the corrosive medium without the peelings of the coatings.

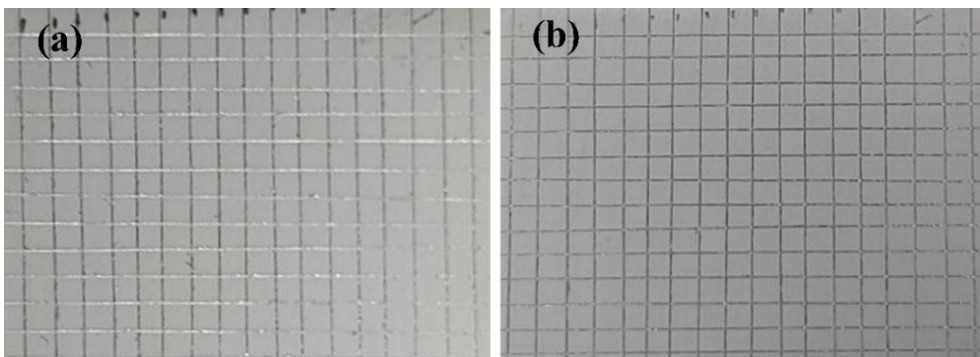

**Figure 9.** The images of the Mg/P/Z/F/H surface before (**a**) and after (**b**) the tape test.

### 3.6. Self-Cleaning Test

The ash adhesion resistance of the coating was tested by using the chalk dust for exploring the self-cleaning properties. The self-cleaning ability of the superhydrophobic Mg/P/Z/F/H composite material is presented in Figure 10. The Mg/P/Z/F/H samples were kept at an inclination of 10°, and 10 µL water droplets were dropped from a height of 20 mm on the area marked in red. The specified area was completely exposed after 10 water droplets, and the water droplets took away the chalk dust fully. However, the water droplets were observed to stay on the surfaces of the Mg, Mg/P, Mg/P/Z, Mg/P/Z/F samples and did not remove the chalk dust.

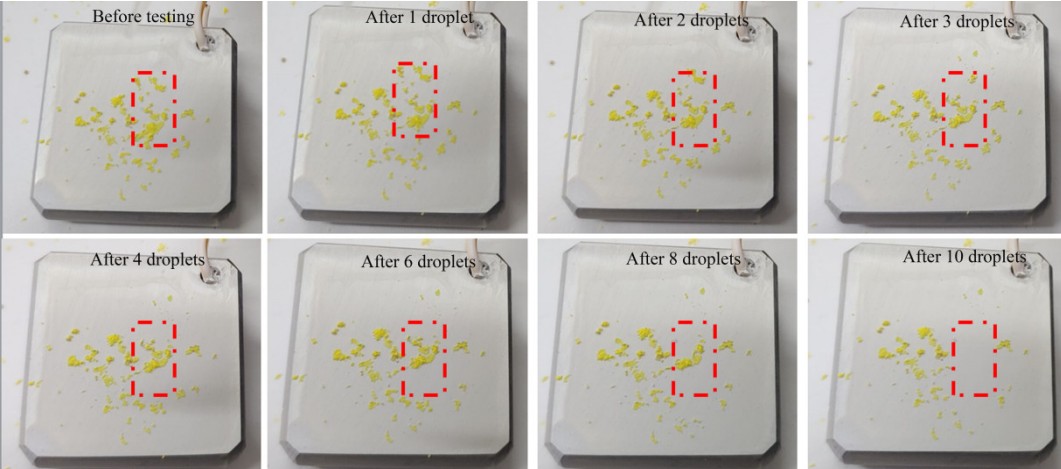

**Figure 10.** The self-cleaning performance of the superhydrophobic M/P/Z/F/H surface.

## 4. Conclusions

In this study, the environmentally degradable superhydrophobic Mg/P/Z/F/H composite materials with less heavy metals were successfully prepared by using the simple dip-coating and heating process. In 3.5 wt.% NaCl, the electrochemical studies manifested in outstanding corrosion resistance of Mg/P/Z/F/H. The heat process at 50 °C for 30 min resulted in the molecules rearrangement, thus, imparting the hydrophobic and superhydrophobic character. Moreover, the Mg/P/Z/F/H material exhibited efficient self-cleaning properties, good adhesion strength, and stability in the wet atmosphere. Overall, the developed materials exhibit high potential for application in construction, automotive, shipbuilding, and medical engineering.

**Author Contributions:** Conceptualization, Z.X. and C.Y.; Formal analysis, C.Y.; Funding acquisition, C.Y. and X.B.; Methodology, Z.X., C.Y., X.B., C.W. and A.N.; Resources, X.B.; Writing—original draft, Z.X.; Writing—review & editing, C.W. and A.N. All authors have read and agreed to the published version of the manuscript.

**Funding:** This research was funded by the National Natural Science Foundation of China (Grant No. 52071246).

**Institutional Review Board Statement:** Not applicable.

**Informed Consent Statement:** Not applicable.

**Data Availability Statement:** Data is contained within the article.

**Conflicts of Interest:** The authors declare no conflict of interest.

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
