# Peer review of "Preparation of Degradable Superhydrophobic Mg/P/Z/F/H Composite Materials and Their Anticorrosion"

_coatings, doi:10.3390/coatings11101239_

Round 1
Reviewer 1 Report
The research and development of environmentally degradable materials is a very interesting research topic regarding the nowadays environmental concerns. In this way, for instance in the introduction section the authors could highlight more the importance of th need of degradable materials towards the application fields with practical examples. The authors can explaining better what are multifunctional surperhydrophobic constructional materials.
Author Response
Respond: Thank you very much for your comments and we revised our manuscript carefully. We added the practical examples in the introduction section. In other parts of our manuscript, the descriptions of our surperhydrophobic materials have been rewrote. In introduction, we added “And the widespread applications of degradable materials can decrease the pollutions of the environment with the abandoned ships in sea, scrapped cars in wild, excess building materials and so on.” In the conclusion, the description was revised as “What’s more, the Mg/P/Z/F/H materials have well self-cleaning properties, good adhesion strength and stability in wet atmosphere and provide a feasible scheme in constructional materials, auto industry, shipbuilding, technology and medical engineering applications.”
Reviewer 2 Report
The manuscript "Preparation of degradable superhydrophobic Mg/P/Z/F/H composite materials and their anticorrosion" is devoted to fabrication of superhydrophobic coatings on magnesium using composite with ZnO nanoparticles as filler, polycaprolactone as adhesive and fluorooxysilane as hydrophobic agent. It was demonstrated that heating of such composite to the temperature just before polycaprolactone melting temperature causes PFDTES molecules rearrange with accompanying increase in hydrophobicity. Manuscript should be improved, before it can be accepted for publishing.
1) Introduction does not contain review of other SHC on magnesium. Correspondingly no comparison of obtained results with presented in the literature is done in discussion section.
2) It is not clear why authors use term “one-step method”, since even if omit P/Z/F preparation samples need to be deep coated and then heated.
3) information about “2956 and 2875” bands is repeated twice.
4) 832 and 1172 bands are not clearly seen in the FTIR results (the features with the same magnitude can be found for other spectra). I suggest to omit this part.
5) Significance of the trend of decreasing contact angles upon exposure to wet atmosphere can not be assessed without errors presented on the graph 8.
6) Since SHC coating on magnesium tend to deteriorate rapidly upon contact with corrosion media for a few hours, it would be nice to add information about corrosion resistance after more prolonged contact with sodium chloride solution. It also would be nice to compare results with corrosion resistance of other SHC cotatings presented in the literature (see point 1)
7) Grammar, especially in the second half of the paper, should be improved: multiple typos (like carves instead of curves), missed spaces etc.
Author Response
Comments and Suggestions for Authors
The manuscript "Preparation of degradable superhydrophobic Mg/P/Z/F/H composite materials and their anticorrosion" is devoted to fabrication of superhydrophobic coatings on magnesium using composite with ZnO nanoparticles as filler, polycaprolactone as adhesive and fluorooxysilane as hydrophobic agent. It was demonstrated that heating of such composite to the temperature just before polycaprolactone melting temperature causes PFDTES molecules rearrange with accompanying increase in hydrophobicity. Manuscript should be improved, before it can be accepted for publishing.
Comment 1:
- Introduction does not contain review of other SHC on magnesium. Correspondingly no comparison of obtained results with presented in the literature is done in discussion section.
Respond: Thank you very much for your comments. The surface of Mg/P/Z/F/H composite materials is superhydrophobic and we hope our coating preparation method can be prepared on different materials’ surfaces. And we prepared the superhydrophobic surface on copper samples. Here we consider that too much review of superhydrophobic coatings on magnesium may limit the reviewers’ understanding of this materials’ application scope.
Comment 2:
- It is not clear why authors use term “one-step method”, since even if omit P/Z/F preparation samples need to be deep coated and then heated.
Respond: Thank you very much for your comments and we revised our manuscript carefully. The descriptions of preparations have been rewrote. In our manuscript, the “one-step method” was revised as “dip-coating method and heating process”.
Comment 3:
- information about “2956 and 2875” bands is repeated twice.
Respond: Thank you very much for your comment and we revised this part.
Comment 4:
- 832 and 1172 bands are not clearly seen in the FTIR results (the features with the same magnitude can be found for other spectra). I suggest to omit this part.
Respond: Thank you very much for your comment. In this part, 832 and 1172 bands of Mg/P/Z/F/H samples can be seen in FTIR which is accordingly to the EDS results. So we hope retain this part.
Comment 5:
- Significance of the trend of decreasing contact angles upon exposure to wet atmosphere can not be assessed without errors presented on the graph 8.
Respond: Thank you very much for your comment and we added the errors in annotation of Fig 8 as “Figure 8. The wet atmosphere of the stability test (a) and the results of stability at wet atmosphere (+1.8o and -1.6o).”.
Comment 6:
- Since SHC coating on magnesium tend to deteriorate rapidly upon contact with corrosion media for a few hours, it would be nice to add information about corrosion resistance after more prolonged contact with sodium chloride solution. It also would be nice to compare results with corrosion resistance of other SHC cotatings presented in the literature (see point 1)
Respond: Thank you very much for your comment. The Tafel extrapolation of the Mg/P/Z/F/H in 3.5 wt. % NaCl for 6h illustrated that the corrosionpotential (Ecorr), corrosioncurrentdensity (icorr) changed slightly. After 6 hours’ immersing in 3.5 wt. % NaCl the Ecorr is about -1.1866 V and the icorr is about 1.1703×10-7A/cm2. The results illustrated that the Mg/P/Z/F/H materials was stable in NaCl corrosive medium. In the study, we added the description about this part.
Comment 7:
7) Grammar, especially in the second half of the paper, should be improved: multiple typos (like carves instead of curves), missed spaces etc.
Respond: Thank you very much for your comments and we revised the manuscript as carefully as we could.

Reviewer 3 Report
The reviewed manuscript has practical relevance, but contains significant methodological issues, which in my opinion require their attention.
Although the meaning and intent of the article can be gleaned with some effort, the manuscript requires attention, both as to its structure and English. I mention some indicative linguistic errors (only those found early on in the manuscript) but there are numerous more throughout it and I’d ask the authors to carefully proofread their manuscript again and consult a colleague versed in scientific English writing:
- Page 1, Line 11, “In this study, the degradable superhydrophobic…” should read “In this study, degradable superhydrophobic…”
- Page 1, Line 13, “…were prepared through one-step method for enhancing…” should read ““…were prepared through a one-step method, for enhancing…”
- Page 1, Lines 16-17, “…composite materials had well self-cleaning…” I believe the word “well” is not the proper choice here, please rephrase.
- Page 1, Line 22-25, The entire first sentence of the manuscript’s introduction is difficult to understand and need re-writing.
- Page 1, Line 25, “And the present superhydrophobic materials …, present where?
- Page 1, Line 26, “…researches focus on the…” is “researches” used as a verb here?
- Page 1, Line 26, “…the undegradable polymer…” using singular here, doesn’t make sense as the parenthesis features 5 indicative matrixes…
Furthermore, the use of abbreviations in the abstract is not appropriate and the abstract should be more descriptive of the employed methodology an results, e.g. how was adhesion and stability evaluated? What does good adhesion mean?
Keywords are requested by journals to ease the identification of a manuscript and are used by search engines, complementary to the title. As such, it makes no sense to use keywords that are repeated in the title.
The preparation procedure of the Mg/P/Z/F/H composite materials is extremely confusing! Based on table 1, I understand that not all composites contained ZnO or PFDTES? Why call all composites Mg/P/Z/F/H?
In table 1, the authors mention “Bare” sample, I believe this should be replaced with “uncoated”. Nevertheless, this sample, which I would consider as the reference material, is not tested e.g. against wetting, why?
Why were the surface characteristics of neither uncoated nor Mg/P samples examined?
Why were self-cleaning properties and stability tested only for the M/P/Z/F/H samples?
Please use the same color coding in figures 3, 5 and 6.
I believe, that without addressing these issues, it is very difficult to assess the quality of the results and discussion of the manuscript.
Author Response
Comments and Suggestions for Authors
The reviewed manuscript has practical relevance, but contains significant methodological issues, which in my opinion require their attention.
Although the meaning and intent of the article can be gleaned with some effort, the manuscript requires attention, both as to its structure and English. I mention some indicative linguistic errors (only those found early on in the manuscript) but there are numerous more throughout it and I’d ask the authors to carefully proofread their manuscript again and consult a colleague versed in scientific English writing:
Comment 1:
Page 1, Line 11, “In this study, the degradable superhydrophobic…” should read “In this study, degradable superhydrophobic…”
Page 1, Line 13, “…were prepared through one-step method for enhancing…” should read ““…were prepared through a one-step method, for enhancing…”
Page 1, Lines 16-17, “…composite materials had well self-cleaning…” I believe the word “well” is not the proper choice here, please rephrase.
Page 1, Line 22-25, The entire first sentence of the manuscript’s introduction is difficult to understand and need re-writing.
Page 1, Line 25, “And the present superhydrophobic materials …, present where?
Page 1, Line 26, “…researches focus on the…” is “researches” used as a verb here?
Page 1, Line 26, “…the undegradable polymer…” using singular here, doesn’t make sense as the parenthesis features 5 indicative matrixes…
Furthermore, the use of abbreviations in the abstract is not appropriate and the abstract should be more descriptive of the employed methodology an results, e.g. how was adhesion and stability evaluated? What does good adhesion mean?
Respond:
Thank you very much for your comments and we revised our manuscript carefully. Page 1, Line 11and Page 1, Line 13, we revised this part to “…were prepared through a one-step method, for enhancing…”
Page 1, Lines 16-17, we revised this part to “…composite materials had efficient self-cleaning…”.
Page 1, Line 22-25, the entire first sentence of the manuscript’s introduction is re- written as “With special solid-liquid adhesion, self-cleaning,[1-11] anti-icing,[4,6,7-11] anti-corrosion,[5,9,12,13] antibacterial and other outstanding characteristics,[14-16] multifunctional super-antiwetting surfaces becomes the research hotspots in the world both in preparations and applications.”
Page 1, Line 25, this part was revised as “And the superhydrophobic materials researches focus on the undegradable polymers”
Page 1, Line 26, the word “research” can be used as noun & verb, here the “research” is a noun. Furthermore in this sentence the “focus” is a verb.
Page 1, Line 26, this part was revised as “With the wide application of metals and the undegradable polymers, the environment risks…”
Comment 2:
Keywords are requested by journals to ease the identification of a manuscript and are used by search engines, complementary to the title. As such, it makes no sense to use keywords that are repeated in the title.
Respond:
Thank you very much and this comment is helpful for us to improve our manuscript. And the keywords were re-wrotten as “superhydrophobic surface; heating process; Dip-coating; poly(ε-caprolactone) (PCL)”.
Comment 3:
The preparation procedure of the Mg/P/Z/F/H composite materials is extremely confusing! Based on table 1, I understand that not all composites contained ZnO or PFDTES? Why call all composites Mg/P/Z/F/H?
Respond:
Thank you very much for your comment about this part. In table 1, four different hydrophobicities of Mg/P, Mg/P/Z, Mg/P/Z/F and Mg/P/Z/F/H were showed and they were all the composite materials. Here we revised this part as “The resulting surfaces showed different hydrophobicities of composite Mg/P (with PCL), Mg/P/Z (with PCL and ZnO), Mg/P/Z/F (with PCL, ZnO and PFDTES) and Mg/P/Z/F/H (with PCL, ZnO and PFDTES) materials”. And in our study, we prepared four composite materials (in Table 1) and tested their wetting behaviors and so on (in part 3. RESULTS AND DISCUSSION).
Comment 4:
In table 1, the authors mention “Bare” sample, I believe this should be replaced with “uncoated”. Nevertheless, this sample, which I would consider as the reference material, is not tested e.g. against wetting, why?
Respond:
Thank you very much for your comment and here we consider that we prepared four different hydrophobicities of Mg/P (with PCL), Mg/P/Z (with PCL and ZnO), Mg/P/Z/F (with PCL, ZnO and PFDTES) and Mg/P/Z/F/H (with PCL, ZnO and PFDTES) materials. We thought twice and revised the description about the “Bare” samples. In our study, the “Bare” magnesium alloys were modified to “Mg” samples. As is known, most of metals are hydrophilic. And the surface of magnesium alloys is hydrophilic as we tested by water droplets. Here in our preparing process of composite materials in dichloromethane (DCM) solvent, we do not consider the water wetting ability of the surface of magnesium alloys. So the figures of magnesium alloys’ wetting properties were not shown in the manuscript.
Comment 5:
Why were the surface characteristics of neither uncoated nor Mg/P samples examined?
Respond:
Thank you for your comment. Here the uncoated samples (“Mg” samples) were tested too much in researches and it is not helpful for the readers to understand the composite Mg/P, Mg/P/Z, Mg/P/Z/F and Mg/P/Z/F/H materials. In our manuscript, the wetting tests showed that the Mg/P is hydrophilic with poor corrosion resistance comparing with Mg/P/Z, Mg/P/Z/F and Mg/P/Z/F/H samples. Here the porous structure of PCL coatings of the Mg/P provides the limited protection to the magnesium. Furthermore, the PCL coatings were discussed in former studies. (Preparation of novel functional Mg/O/PCL/ZnO composite biomaterials and their corrosion resistance, Applied Surface Science, 351 (2015) 410-415.). In this study we added this part and discussed the surface characteristics of the PCL coatings of Mg/P as “The Mg/P is with high porosity it is not enough to effectively protect magnesium alloys substrates against the corrosive medium rely on the high porous PCL coatings of Mg/P.[25].”
Comment 6:
Why were self-cleaning properties and stability tested only for the M/P/Z/F/H samples?
Respond:
Thank you very much for your comment. In general, superhydrophilic coatings can exhibit the ability to remove the hydrophobic contaminants including oil from the coating surface in aqueous media, which is regarded as the self-cleaning property. And in our study, the tests illustrated that the water droplets could not remove the chalk ash on the surface of the Bare, Mg/P, Mg/P/Z and Mg/P/Z/F samples. The self-cleaning ability is the unique property of superhydrophobic surfaces. And we display self-cleaning test of the M/P/Z/F/H samples in our manuscript.
Mg |
Mg/P |
Mg/P/Z |
Mg/P/Z/F |
|
|
|
|
Comment 7:
Please use the same color coding in figures 3, 5 and 6.
Respond:
Thank you very much for your comment. The colors of lines in Figs have been revised.

Round 2
Reviewer 2 Report
Not all reviewer comments were considered appropriately, thus I suggest minor revision.
In particular,
comment 1. It is critical for any study to locate itself in current state of the field and to be able to compare results to other presented in the literature. Thus still at least some comparison with SHC on magnesium is required.
Comment 5. In my comment for figure 8(b), I meant error bars on the graph since for now it is impossible distinguish qualitative change and statistical variance.
Author Response
Comments and Suggestions for Authors
Not all reviewer comments were considered appropriately, thus I suggest minor revision.
In particular,
comment 1. It is critical for any study to locate itself in current state of the field and to be able to compare results to other presented in the literature. Thus still at least some comparison with SHC on magnesium is required.
Respond:Thank you very much for your comment and the comparison with SHC on magnesium were added in the part of introduction (Compared with the plasma electrolytic oxidation (PEO) [39,40], hydrothermal treatment [41] and electrodeposition [42] techniques, On the other hand, in environ-mentthe non-degradable polymer coatings on the magnesium alloys, such as polypro-pylene [43], polyvinyl chloride [44] and epoxy resin [45], cannot degrade and may bring the potential harmful effects on the organisms.) Further, in introduction this part about the preparation of Mg/P/Z/F/H was revised (The rough Mg/P/Z/F/H structure was constructed by using the dip-coating method to effectively protect the magnesium alloy. Furthermore, the heating process for 30 minutes at 50oC was applied to repair the defects on the surface of the composite Mg/P/Z/F materials as well as to rearrange the PCL and PFDTES molecules to transform the composite surface from hydrophobic (96.5o) to superhydrophobic (159.0o).). In the end, the conclusion was partly modified as “In this study, the environmentally degradable superhydrophobic Mg/P/Z/F/H composite materials with less heavy metals were successfully prepared by using the simple dip-coating and heating process. And in 3.5 wt.% NaCl the electrochemical studies manifested the outstanding corrosion resistance of Mg/P/Z/F/H. The heat process at 50oC for 30 minutes resulted in the molecules rearrangement, thus, imparting the hydrophobic and superhydrophobic character. Moreover, the Mg/P/Z/F/H material exhibited efficient self-cleaning properties, good adhesion strength and stability in the wet atmosphere. Overall, the developed materials exhibit high potential of application in construction, automotive, shipbuilding and medical engineering fields.”.
Comment 5. In my comment for figure 8(b), I meant error bars on the graph since for now it is impossible distinguish qualitative change and statistical variance.
Respond:We deeply appreciate your comment and redraw figure 8(b) with error bars.

Reviewer 3 Report
Although the authors have addresses most of my points, linguistic errors persist throughout the manuscript. Only the ones directly indicated by me were rectified and it doesn't seem like the authors proof read the manuscript or consulted with a native speaker.
Author Response
Comments and Suggestions for Authors
Although the authors have addresses most of my points, linguistic errors persist throughout the manuscript. Only the ones directly indicated by me were rectified and it doesn't seem like the authors proof read the manuscript or consulted with a native speaker.
Respond:Thank you very much for your comments and the manuscript was with attentive proofreading & proper copyediting by professional organization (Editsprings).
In introduction, “…[14-16] multifunctional super-antiwetting surfaces are among the research hotspots in the world both in preparations and applications.” Was revised as “…[14-16], the multifunctional super-antiwetting surfaces have received an extensive research attention with respect to preparations and applications.” “As one of the lightest metals, with a density two-thirds that of aluminium and one-quarter that of steel, magnesium alloys thus have the great potential to improve system performance and energy efficiency in aerospace, auto industry, shipbuilding, mobile electronics, and bioengineering applications because of the magnesium’s excel-lent chemical, mechanical, biological and physical properties.[23-27] ” was modified as “As one of the lightest metals, with density two-third that of aluminium and one-quarter that of steel, the magnesium alloys exhibit a significant potential to improve the system performance and energy efficiency in aerospace, automotive, shipbuilding, mobile electronics and bioengineering applications owing to the excellent chemical, mechanical, biological and physical properties of magnesium [23-27].” “ZnO powders are used in electronic and optoelectronic devices, solar cells and pharmaceutical engineering because of the high safety, low price, extraodinary opto-electronic property and lacking of polluting effects as a newer type of promising candidate.[35,37,38]” was revised as “The ZnO powder is used in the electronic and optoelectronic devices, solar cells and pharmaceutical applications due to the high safety, low price, extraodinary opto-electronic characteristics and non-polluting character [35,37,38].”
There are other modifications and collation in parts of experimental section & results and discussion such as “Under magnetic stirring for 5 h, PCL (5 wt.%) granules were dissolved in 60mL di-chloromethane (DCM) solvent. Then mixed the ZnO powders with PCL polymer solu-tion (5 wt.%) and stirred continuously. Dropt 1.5 mL PFDTES and stirred continuously for 10 h. The prepared samples were immersed into the mixed solutions for 30 s and pulled out of the solution at a speed of 2 mm/s. Then the Mg/P/Z/F composite materials were prepared.” was modified as “The PCL (5 wt.%) granules were dissolved in 60 mL dichloromethane (DCM) under magnetic stirring for 5 h. Subsequently, the ZnO powder was added to the PCL solution (5 wt.%) and stirred continuously. It was followed by the addition of 1.5 mL PFDTES and continuous stirring for 10 h. The prepared samples were immersed in the mixed solutions for 30 s and pulled out of the solution at a speed of 2 mm/s, leading to the formation of the Mg/P/Z/F composite materials.” “The SEM and EDS surface morphology of M/P/Z, M/P/Z/F and M/P/Z/F/H compo-site materials were shown in Fig 2. The Mg/P is with high porosity and it is not enough to effectively protect magnesium alloys substrates against the corrosive medium rely on the high porous PCL coatings of Mg/P.[25]” was revised as “The SEM and EDS analyses of Mg/P/Z, Mg/P/Z/F and Mg/P/Z/F/H are presented in Fig 2. Mg/P exhibited high porosity, thus, leading to an ineffective protection of the magnesium alloy substrates against the corrosive medium [25].” “EIS results of Mg/P/Z/F/H samples indicated that after heating process the superhydrophobic samples Mg/P/Z/F/H samples could have desirable anti-corrosion performance.” was changed to “The EIS analysis of Mg/P/Z/F/H indicated that the superhydrophobic surface after the heating process led to a desirable anti-corrosion performance.” “By the Cassie-Baxter state, the air trapped of the micro/nanostructured surface can improve the hydrophobicity and corrosion resistance of Mg/P/Z/F/H materials.[ 47,48] The Mg/P/Z/F/H materials had the best corrosion resistance and the corrosion rate of Mg/P/Z/F/H was 1.9152×10-3 mm/y. Here we found that with the addition of ZnO the corrosion resistance of Mg/P/Z sample decreased obviously because the ZnO powders destroyed the continuity of PCL layer (Fig 2d). And with the mixing of PFDTES the layer’s discontinuity was modified (Fig 2e). Furthermore, the Mg/P/Z/F/H materials’ main component parts are PCL, ZnO and magnesium alloys with the degradation and biocompatibility [53-56].” was revised as “Based on the Cassie-Baxter state, the air trapped on the micro/nanostructured surface improved the hydrophobicity and corrosion resistance of Mg/P/Z/F/H [47, 48]. Mg/P/Z/F/H exhibited the best corrosion resistance, with the corrosion rate determined to be 1.9152×10-3 mm/y. The addition of ZnO obviously decreased the corrosion resistance of Mg/P/Z as the ZnO powder disturbed the continuity of the PCL layer (Fig 2d). In addition, the discontinuity of the layer was modified after mixing PFDTES (Fig 2e). Furthermore, the main components of Mg/P/Z/F/H are PCL, ZnO and magnesium alloys with the degradation and biocompatibility [53-56].” “In this study, environmentally degradable superhydrophobic Mg/P/Z/F/H compo-site materials with less heavy metals were prepared successfully by simple dip-coating and heating process. And in 3.5 wt.% NaCl the electrochemical studies manifested that the Mg/P/Z/F/H had the outstanding corrosion resistance. About process of the super-hydrophobic Mg/P/Z/F/H at 50oC for 30 minutes, we proposed the molecules rearrangement mechanism to explain the hydrophobic and superhydrophobic change process. What’s more, the Mg/P/Z/F/H materials have efficient self-cleaning properties, good adhesion strength and stability in wet atmosphere and provide a feasible scheme in constructional materials, auto industry, shipbuilding, technology and medical engineering applications.” was modified as “In this study, the environmentally degradable superhydrophobic Mg/P/Z/F/H composite materials with less heavy metals were successfully prepared by using the simple dip-coating and heating process. And in 3.5 wt.% NaCl the electrochemical studies manifested the outstanding corrosion resistance of Mg/P/Z/F/H. The heat process at 50oC for 30 minutes resulted in the molecules rearrangement, thus, imparting the hydrophobic and superhydrophobic character. Moreover, the Mg/P/Z/F/H material exhibited efficient self-cleaning properties, good adhesion strength and stability in the wet atmosphere. Overall, the developed materials exhibit high potential of application in construction, automotive, shipbuilding and medical engineering fields.”

This manuscript is a resubmission of an earlier submission. The following is a list of the peer review reports and author responses from that submission.
Round 1
Reviewer 1 Report
The magnesium alloys are the utmost importance in many applicaiton fileds, however the corrosion resistence is the major problem limiting their application. The work presented by authors is very actual and the superhydrophobic layer is a environmentally friendly solution revealing the need of development of new coating solutions based on "green chemistry".
Regarding the dip-coating method, the thickness of the coating/layer is manly defined by the withdrawal speed, viscosity of the liquid. The authors did not mention the thickness of the as-prepared layers in the manuscript. Is it possible to estimate the thickness of the layers? Since the dip-coating method, usually produce a thickness gradient. The thickness plays an important role in the corrosion proctection. For instance, the porosity of the layers can be assessed by means of electrochemical measurements, polarization resistances, these values can provide more information about the performance of the layers.
Author Response
Comments and Suggestions for Authors
The magnesium alloys are the utmost importance in many applicaiton fileds, however the corrosion resistence is the major problem limiting their application. The work presented by authors is very actual and the superhydrophobic layer is a environmentally friendly solution revealing the need of development of new coating solutions based on "green chemistry".
Regarding the dip-coating method, the thickness of the coating/layer is manly defined by the withdrawal speed, viscosity of the liquid. The authors did not mention the thickness of the as-prepared layers in the manuscript. Is it possible to estimate the thickness of the layers? Since the dip-coating method, usually produce a thickness gradient. The thickness plays an important role in the corrosion proctection. For instance, the porosity of the layers can be assessed by means of electrochemical measurements, polarization resistances, these values can provide more information about the performance of the layers.
Answers:Thank you very much for your Comments. In our article,we demonstrated that the As a widely applied surface treatment technology, dip-coating can produce a relatively adherent, stable and uniform films on the surface of materials.[39-42] And here by doping only 1.5mL PFDTES to the PCL polymer (5 wt.%) and ZnO powders (5 wt.%) composite solution (60mL) could not change the viscosity of the liquid too much. More essentials part in our article is that without change of the thickness of the composite film of Mg/P/Z/F materials and just through the heating process at 50oC for 30 min, the CA of Mg/P/Z/F sample is from hydrophobic (96.5±1.8o) to superhydrophobic (159.0±1.6o). Furthermore, improvement of the hydrophobicity from Mg/P/Z/F (CA~96.5±1.8o) to Mg/P/Z/F/H (CA~159.0±1.6o) could strengthen the anticorrosion resistance of the composite materials (Fig 6 and Table 2).

Reviewer 2 Report
Review of : Preparation of environmentally degradable superhydrophobic 2 Mg/P/Z/F/H composite materials
Even though the topic has high interest, then English is horrible. The English level is at a very low level and do not help in understanding the manuscript.
Other issues are below:
The PTC is not well explained, its chemistry and the corresponding chemical reaction if any.
The water contact angle is not described well how measured.
It is not clear why the surface become super hydrophilic, is it due to the chemical reaction, the surface properties??
The model showing molecules rearrangement is not clear, and no proof was provided
The corrosion test is not well explained and why the Nyquist plane is a good fit for explaining it.
The air trapped was not proven or shown experimentally.
The adhesion and Scribe Grid was not explained as well as its suitability.
The self cleaning test is not clear.
Author Response
Comments and Suggestions for Authors
Review of : Preparation of environmentally degradable superhydrophobic 2 Mg/P/Z/F/H composite materials
Even though the topic has high interest, then English is horrible. The English level is at a very low level and do not help in understanding the manuscript.
Answer: We modified our manuscript and thank you for your kind comments.
Other issues are below:
The PTC is not well explained, its chemistry and the corresponding chemical reaction if any.
Answer: PCL() is stable at low temperature(no higher than 60oC). Here the PCL plays a part as the adhesive to combine the magnesium sample and ZnO powders. The ZnO and 1H,1H,2H,2H-perfluorodecyltriethoxysilane (PFDTES) can form chemical bond (,ACS Appl. Mater. Interfaces. 2018, 10, 13452-13461). In our study, Zn-O-Si bond is not the point to change the water contact angle (before and after heating process at 50oC). The reason about the change of CA is the molecules rearrangement to expose the PFDTES molecules.
It is not clear why the surface become super hydrophilic, is it due to the chemical reaction, the surface properties??
Answer: In our article the surface become superhydrophilic is not due to the chemical reaction. After 30minutes’ heating progress, the PFDTES and PCL molecules cause molecules rearrangement and the PFDTES molecules are exposed on the surface of the composite M/P/Z/F/H materials to form the superhydrophobic surface (Fig 4). The PCL is stable at low temperature in the air.
The model showing molecules rearrangement is not clear, and no proof was provided
Answer: Firstly, the EDS results showed that the contents of PFDTES on the surface of composite materials increased obviously after heating process at 50oC. Secondly,at 832 cm−1 for Zn-O stretching for Mg/P/Z/F/H could explain that after heating process the PCL transfer occurs on ZnO. Before heating process the PCL molecules coatings completely cover the ZnO powders, and the Zn-O stretching was not detected. After heating process the PCL molecules transferred on the surface of ZnO and the Zn-O stretching was detected (FTIR). In the SEM of Mg/P/Z/F (h) and Mg/P/Z/F/H (i), the repaired gully defects of the surface provide the evidence of the change on the surface of composite materials after heating process. Then, the change of hydrophobicity was matching with the change of PCL molecules on the surface of composite materials from hydrophobic to superhydrphobic.
The corrosion test is not well explained and why the Nyquist plane is a good fit for explaining it.
Answer: The Nyquist plane can provide the essential information of the charge transfer resistance (Rct). The diameter of the capacitive loop in the high frequency range represented the charge transfer resistance (Rct). The Rct value reflected how difficult the electrochemical corrosion reaction might be. The higher Rct value means the coatings has higher corrosion resistance. (Journal of Alloys and Compounds. 2009, 488, 392-399& J. Appl. Electrochem. 2006, 36, 195-204).
The air trapped was not proven or shown experimentally.
Answer: The air trapped is proven by its low sliding angle and our self-cleaning test showed the state of the surperhydrophobicty was Cassie-Baxter state. At low angle the water droplets slide easily in our experimental tests (Self-cleaning test, the water droplet can easily move at low sliding angle). The Cassie-Baxter state is with low sliding angle because of the air trapped. The Cassie-Baxter and Wenzel state are the two different superhydrophobic surfaces.
The Cassie-Baxter state has the air trapped with a low sliding angle. The Wenzel state does not have the air trapped and the water droplet cannot slide easily. (Colloids and Surfaces A: Physicochemical and Engineering Aspects. 2021, 625, 126927& ACS Applied Materials & Interfaces. 2012, 4, 8, 4348-4356) The water droplet could slide easily at low angle on the surface of the superhydrophobic composite materials in our study and the self-cleaning test showed that the superhydrophobic surface was the Cassie-Boxter state.
The adhesion and Scribe Grid was not explained as well as its suitability.
Answer: The Scribe Grid test is the widely used method to measure the adhesion of coating and substrates The Scribe Grid test methods cover procedures for assessing the adhesion of relatively ductile coating films to metallic substrates by applying and removing pressure-sensitive tape over cuts made in the film. These test methods are used to evaluate whether the adhesion of a coating to a substrate is adequate for the user’s application. They do not distinguish between higher levels of adhesion for which more sophisticated methods of measurement are required. This test method is similar in content (but not technically equivalent) to ISO 2409.
The self cleaning test is not clear.
Answer: The self-cleaning test is as described in part 3.6(Self-cleaning test). The Mg/P/Z/F/H samples were kept at an inclination of 10°. Each time 10μL water droplet was dropped to the red specified area. The water droplet rolled off along the sample with the chalk dust immediately. After ten water droplets, the red specified area was completely clean without chalk dust. The self-cleaning test illustrated that in water environment the surface of superhydrophobic composite mateterials could not be polluted by dust easily. Thank you very much for your suggestions and comments.

Reviewer 3 Report
In this manuscript, the authors address the preparation of superhydrophobic Mg/P/Z/F/H composite materials. Different electrochemical and surface characterization methods were used. However, I found the paper unclear and think that the novelty is not fully addressed.
The introduction is very summarized and provides insufficient background. Also the novelty of the study is not clearly demonstrated, what this study brings differently from what has been made so far?
The explanation of experiments is too poor, as well as the results discussion. Furthermore, the structure of the results/discussion section is confused, e.g. in the section "3.1. Wetting behaviors" the authors mainly comment the contact angle values, without explaining or discuss why they change; part of this explanation appears in a further section (3.4. Interface model and anticorrosion), but it seems out of context.
In general, none of the results presented are discussed in detail.
Moreover the title of the manuscript says "Preparation of environmentally degradable superhydrophobic composite materials" and no tests regarding degradation or eco toxicity are present in the paper. Is briefly mentioned in the introduction that the constituents of these composites are biodegradable and/or non-toxic by their own, however toxicity or degradation of the final system is not presented. AZD91 is not considered a biodegradable alloy; also the silane used (PFDTES) raises concerns in terms of toxicity. So a screening of the toxicity and degradability of the final system should have been conducted.
For these reasons, I came to believe that this paper should be rejected.
Author Response
omments and Suggestions for Authors
In this manuscript, the authors address the preparation of superhydrophobic Mg/P/Z/F/H composite materials. Different electrochemical and surface characterization methods were used. However, I found the paper unclear and think that the novelty is not fully addressed.
The introduction is very summarized and provides insufficient background. Also the novelty of the study is not clearly demonstrated, what this study brings differently from what has been made so far?
The explanation of experiments is too poor, as well as the results discussion. Furthermore, the structure of the results/discussion section is confused, e.g. in the section "3.1. Wetting behaviors" the authors mainly comment the contact angle values, without explaining or discuss why they change; part of this explanation appears in a further section (3.4. Interface model and anticorrosion), but it seems out of context.
In general, none of the results presented are discussed in detail.
Moreover the title of the manuscript says "Preparation of environmentally degradable superhydrophobic composite materials" and no tests regarding degradation or eco toxicity are present in the paper. Is briefly mentioned in the introduction that the constituents of these composites are biodegradable and/or non-toxic by their own, however toxicity or degradation of the final system is not presented. AZD91 is not considered a biodegradable alloy; also the silane used (PFDTES) raises concerns in terms of toxicity. So a screening of the toxicity and degradability of the final system should have been conducted.
For these reasons, I came to believe that this paper should be rejected.
Answers: Thank you for your sincere comments and suggestions.
The novelty of the study was illustrated in our introduction. Firstly, the degradable, ecofriendly and superhydrophobic M/P/Z/F/H composite materials were prepared through one-step method to solve the poor corrosion resistance of magnesium. Then, the M/P/Z/F/H composite materials did not release heavy metal ion in corrosive medium. Furthermore, 30minutes’ heating process at 50oC can not only repair defects of the composite Mg/P/Z/F materials’ surface but also rearrange the PCL and PFDTES molecules to make the composite surface from hydrophobic (96.5o) to superhydrophobic (159.0 o).
About the back ground, in order to reduce negative impact of the undegradable polymer and heavy metal ion release to the water environment. And this study is different from the former researches (And the present superhydrophobic materials researches focus on the undegradable polymer (epoxy resin, polyurethane, polytetrafluoroethylene , polyvinyl chloride, perfluoropolyalkylether. etc) and various metal alloys (Fe, Cu, Ni, Co) containing heavy metal ions.). Then, the corrosion resistance of the composite materials can be increased to satisfied the engineering technology and medical engineering application (Introduction & Conclusion).
In the section “3.1. Wetting behaviors” and “3.4. Interface model and anticorrosion”, “The simplest and most direct measure of surface wettability is the contact angle (CA) of a hemispherical liquid droplet (often a water droplet) on the surface of interest.”(Anisotropic Wetting Surfaces with One-Dimesional and Directional Structures: Fabrication Approaches, Wetting Properties and Potential Applications, Adv. Mater. 2012, 24, 1287-1302) In section 3.4, we added the description that “the air film can sparate the substrate from the corrosive medium to improve the corrosion resistance”.
The reviewer point out that there is “no tests regarding degradation or eco toxicity are present in the paper”. Firstly, the AZ91D, ZnO and PCL are all the degradable and eco-friendly materials. There are many researches about these materials’ toxicity and the related cited articles all remind it. Then, the AZ91D magnesium is biodegradable magnesium alloys without seriously obvious toxicity. (Biodegradable Magnesium Alloys Promote Angio-Osteogenesis to Enhance Bone Repair, Adv. Sci. 2020, 7, 2000800; 1992 Toxicity of Metals (ASM Metals Handbook vol 2) (Materials Park, OH : ASM International) chap Special Engineering Topics); Mg alloys can release bio-functional metallic ions that promote the nerve repair. (Bone-like matrix formation on magnesium and magnesium alloys, J. Mater. Sci.: Mater. Med. 2008, 19, 407-415)
About the toxicity of PFDTES: PFDTES, as a surface modifier, was not the main components of the materials. And in our study the main body of the composite materials was degradable and ecofriendly. What’s more, the toxicity and degradability of the composite materials are our next research.

Reviewer 4 Report
The manuscript "Preparation of environmentally degradable superhydrophobic Mg/P/Z/F/H composite materials" is dedicated to fabrication of superhydrophobic coatings on magnesium using ZnO nanoparticles to provide developed morphology, polycaprolactone to bind nanoparticles to the surface and fluorooxysilane as hydrophobic agent. It was shown that proposed coating demonstrates corrosion resistance. It was also shown that under quite low heating, surface molecules rearrange and fluorooxysilane moves to outer layer. However, manuscript contains a few flaws which should be revised, before manuscript can be published. Therefore I suggest major revision.
1) First of all, manuscript use “environmentally degradable” in the title, while no study of degradation process is present. In contrast, manuscript is tending to emphasize that proposed coatings are resistant (non-degradable). I suggest to revise manuscript title to match the content.
2) Literature review is quite topic-broad to the detriment of review of advances in creating corrosion-resistant superhydrophobic coatings on magnesium. There are a few promising approaches in that field including PEO, laser texturing or the same approach which authors use (binding functionalized nanoparticles to the surface) which should be presented in the review. See for instance 10.1016/j.jma.2018.02.001, 10.1021/acsomega.7b01256, 10.1680/si.13.00001, 10.1016/j.jmst.2020.02.055, 10.1016/j.matlet.2017.01.050.
3) While some used materials are indeed safe for environment, 1H,1H,2H,2H-perfluorodecyltriethoxysilane is not. Usually to increase safety of coatings which use fluorooxysilanes, treatment includes special polymerization step, when under heat around 130 °C fluorooxysilane molecules bind to each other and form a cross-linked layer which emanate fluorocarbon chemicals to the environment much slower. Statement about “ecofriendly” should be revised.
4) The misleading phrase: “PFDTES molecules to make the compo-site surface from hydrophilic (96.5°) to superhydrophobic (159.0°)”. Contact angle 96.5° does not correspond to hydrophilicity.
5) FTIR data causes a lot of questions:
a) proposed coating should not be IR transparent, how transmittance was measured?
b) the assignment of vibration frequencies should be taken from chemical handbooks
c) on figure 3 there are not clearly shown characteristic peaks (in read circles). Thus this figure does not support statement about “obviously strengthened” CF3 bands (the same problem with ZnO band).
6) EIS spectra causes a few questions as well
a) how long coatings were equilibrated in contact with NaCl solution before EIS was measured?
b) how corrosion resistance (current and potential) change in time during prolonged contact with corrosion media? Note, that only that data truly characterizes the durability, not instantaneous value of corrosion currents.
c) why Mg/P/Z coating demonstrates worse corrosion resistant than Mg/P? Does galvanic pares play role here?
7) The data in Figure 8 without error bars does not make much sense.
8) Using Scribe-Grid Test (ASTM D 3359-78) for this kind of coating is not substantiated: this test shows adhesion of coating to the surface rather than mechanical stability. Sand abrasion tests will be more appropriate. Besides, reference [25] to the another paper when this test was used without description should be changed to the reference to ASTM D 3359-78 itself.
9) Last but no least grammar of the paper should be improve.
Author Response
Comments and Suggestions for Authors
The manuscript "Preparation of environmentally degradable superhydrophobic Mg/P/Z/F/H composite materials" is dedicated to fabrication of superhydrophobic coatings on magnesium using ZnO nanoparticles to provide developed morphology, polycaprolactone to bind nanoparticles to the surface and fluorooxysilane as hydrophobic agent. It was shown that proposed coating demonstrates corrosion resistance. It was also shown that under quite low heating, surface molecules rearrange and fluorooxysilane moves to outer layer. However, manuscript contains a few flaws which should be revised, before manuscript can be published. Therefore I suggest major revision.
1) First of all, manuscript use “environmentally degradable” in the title, while no study of degradation process is present. In contrast, manuscript is tending to emphasize that proposed coatings are resistant (non-degradable). I suggest to revise manuscript title to match the content.
Answer: In our study, the environmentally degradable ZnO, magnesium alloys and PCL are the main body of the composite materials. Then we consider that the title should revise to “Preparation of environmentally degradable superhydrophobic Mg/P/Z/F/H composite materials and their anticorrosion” as the reviewer’s suggestions.
2) Literature review is quite topic-broad to the detriment of review of advances in creating corrosion-resistant superhydrophobic coatings on magnesium. There are a few promising approaches in that field including PEO, laser texturing or the same approach which authors use (binding functionalized nanoparticles to the surface) which should be presented in the review. See for instance 10.1016/j.jma.2018.02.001, 10.1021/acsomega.7b01256, 10.1680/si.13.00001, 10.1016/j.jmst.2020.02.055, 10.1016/j.matlet.2017.01.050.
Answer: This study is focus on the preparation of the degradable and anti-corrosive Mg/P/Z/F/H composite materials not the preparation method. And we revised the main part of our article to satisfy the corrosion resistance researches.
3) While some used materials are indeed safe for environment, 1H,1H,2H,2H-perfluorodecyltriethoxysilane is not. Usually to increase safety of coatings which use fluorooxysilanes, treatment includes special polymerization step, when under heat around 130 °C fluorooxysilane molecules bind to each other and form a cross-linked layer which emanate fluorocarbon chemicals to the environment much slower. Statement about “ecofriendly” should be revised.
Answer: PFDTES, as a surface modifier, was not the main components of the composite materials. And in our study the main body of the composite materials was degradable and ecofriendly. This study focused on the stability at normal atmospheric temperature and working condition (no higher than 80oC).
4) The misleading phrase: “PFDTES molecules to make the compo-site surface from hydrophilic (96.5°) to superhydrophobic (159.0°)”. Contact angle 96.5° does not correspond to hydrophilicity.
Answer: We revised “hydrophilic” to “hydrophobic”. Furthermore, 30minutes’ heating process at 50oC can not only repair defects of the composite Mg/P/Z/F materials’ surface but also rearrange the PCL and PFDTES molecules to make the composite surface from hydrophobic (96.5o) to superhydrophobic (159.0o).
5) FTIR data causes a lot of questions:
- a) proposed coating should not be IR transparent, how transmittance was measured?
Answer: The composite coatings were translucent and could be measured by IR transparent.
- b) the assignment of vibration frequencies should be taken from chemical handbooks
Answer: The assignment of vibration frequencies were not the fixed value and we changed our reference ([44] Martins,P.; Lopes, A.C.; Lanceros-Mendez, S. Electroactive phases of poly (vinylidene fluoride): determination, processing and applications, Progress in Polymer Science,39 (2014) 683-706.).
- c) on figure 3 there are not clearly shown characteristic peaks (in read circles). Thus this figure does not support statement about “obviously strengthened” CF3 bands (the same problem with ZnO band).
Answer: The Fig 3 has been modified and we added the enlarged view. Some expressions were revised as the review’s suggestions to make the explainations more accurate.
6) EIS spectra causes a few questions as well
- a) how long coatings were equilibrated in contact with NaCl solution before EIS was measured?
Answer: In this article, the samples were immersed into NaCl solution for 50min before EIS in order to measure the early corrosion performance.
- b) how corrosion resistance (current and potential) change in time during prolonged contact with corrosion media? Note, that only that data truly characterizes the durability, not instantaneous value of corrosion currents.
Answer: The Mg/P/Z/F/H composite materials are degradable and the process of corrosion in long term is complex with the degradation of the composite materials. Here we mainly explain the anticorrosion at early term in 3.5%wt NaCl.
- c) why Mg/P/Z coating demonstrates worse corrosion resistant than Mg/P? Does galvanic pares play role here?
Answer: The reasons were added in our study. (Section 3.3 Here we found that with the addition of ZnO the corrosion resistance of Mg/P/Z sample decreased obviously because the ZnO powders destroyed the continuity of PCL layer (Fig 2d). And with the mixing of PFDTES the layer’s discontinuity was modified (Fig 2e).)
7) The data in Figure 8 without error bars does not make much sense.
Answer: In Fig 8, the date was the average level of three parallel samples’ tests and each sample was tested at 5 regular points during the same test cycle.
8) Using Scribe-Grid Test (ASTM D 3359-78) for this kind of coating is not substantiated: this test shows adhesion of coating to the surface rather than mechanical stability. Sand abrasion tests will be more appropriate. Besides, reference [25] to the another paper when this test was used without description should be changed to the reference to ASTM D 3359-78 itself.
Answer: We used Scribe-Grid Test (ASTM D 3359-78) to measure the mechanical adhesion stability. In section 3.5, we revised this part.
9) Last but no least grammar of the paper should be improve.
Answer: Thank you very much for your kind suggestions and we revised the main part of our article.

Round 2
Reviewer 3 Report
Seems that the authors made an effort to improve the manuscript, however, there are still many issues that need to be addressed.First of all, an extensive editing of the language and style are required; the authors need to revise carefully the texts and grammar, and correct the typos. Furthermore, the authors need to be careful with the references they cite, I did not check every single of them, but one was brought to my attention in the following sentence: “Furthermore, containing rare heavy metal elements such as Cu, Ni and Pb the magnesium alloys were environment friendly and used as the biodegradable metals.[25] “ – First, the mentioned reference does not match with what is written; secondly the sentence does not make sense, if the alloys contain rare heavy metals, they cannot be considered environmental friendly, and I am mostly certain that cannot be considered biodegradable as well.
The authors state that: “In this study, degradable, ecofriendly and superhydrophobic M/P/Z/F/H composite materials were prepared through one-step method to solve the poor corrosion resistance of magnesium.“ Once again I have reserves regarding this ecofriendly statement. The composite materials developed in this study were not tested with respect to their degradation and ecotoxicity. Although the authors have replied to my former comments, that those studies will be the next research purpose, they are not addressed in the present manuscript, and for that I advise the authors to be careful when they state that the composites are degradable and ecofriendly. Even if some of the constituents of the composites are biodegradable and ecofriendly by their own, do not mean that the same will be applied to the final system. Moreover AZ91 alloy possesses a high amount of aluminum, an element that raises concerns about its safety for human health, and the silane used as a surface modifier, even in a small amount could induce toxicity.
Regarding results section, the EIS measurements need to be discussed in more detail (time constants, physical meaning of those…). Since the authors decided to include “anticorrosion” in the manuscript title, the results section regarding this anticorrosion behavior should be deeper explored and explained. Moreover, the Nyquist graphs presented are not orthonormal, the time of immersion for which results are presented is not mentioned, the Rct values do not present units of measure.
About the polarization curves: can we assess the corrosion rate of Mg alloys by means of Tafel fitting of potentiodynamic polarization data? Is this logical at all? Please consult Progress in Materials Science 89 (2017) 92-193, section 2.1.1. Also the detailed information for Tafel fitting should be listed, such as the selected cathode region and computational method.
In table 2 the units of corrosion rate are not correct, should be (mm/y) not (mm/a).
Also in this section (3.3 Electrochemical corrosion behavior) the authors reach several conclusions, can you explain in detail how, from the polarization curves, it is possible to conclude that “(ii) the addition of PFDTES could improve the uniformity of the complex film and reduce porosity;” and “ (iii) the heating progress of Mg/P/Z/F/H materials could repair the gully defects and achieve molecular rearrangement. “
In the conclusions section is stated “In this study, completely degradable, ecofriendly and superhydrophobic Mg/P/Z/F/H composite materials were prepared successfully.” The results present can prove the preparation of the composites as well as their super hydrophobicity, but none of them gives any information regarding degradation and/or ecotoxicity. Correct the sentence or include results proving what is stated.
Author Response
Comments and Suggestions for Authors
Seems that the authors made an effort to improve the manuscript, however, there are still many issues that need to be addressed.
First of all, an extensive editing of the language and style are required; the authors need to revise carefully the texts and grammar, and correct the typos. Furthermore, the authors need to be careful with the references they cite, I did not check every single of them, but one was brought to my attention in the following sentence: “Furthermore, containing rare heavy metal elements such as Cu, Ni and Pb the magnesium alloys were environment friendly and used as the biodegradable metals.[25] “ – First, the mentioned reference does not match with what is written; secondly the sentence does not make sense, if the alloys contain rare heavy metals, they cannot be considered environmental friendly, and I am mostly certain that cannot be considered biodegradable as well.
Answer: [25] In this work, die-casted Mg alloy (AZ91) with a chemical composition (wt.) of 8.77% Al, 0.74% Zn, 0.18% Mn, 90.31% Mg was used for investigation. There is no heavy metal elements such as Cu, Ni and Pb. Here the “containing rare heavy metal” means “almost no containing heavy metal”. The biodegradations of AZ91 magnesium have done many researches in different systems. AZ magnesium alloys showed low hemolysis rates and good biocompatibility (ACS Appl. Bio Mater. 2020, 3, 531-538 & Thin Solid Films. 2010, 518, 7563-7567& J. Mater. Sci. Technol. 2012, 28, 3, 261-267)
The authors state that: “In this study, degradable, ecofriendly and superhydrophobic M/P/Z/F/H composite materials were prepared through one-step method to solve the poor corrosion resistance of magnesium.“ Once again I have reserves regarding this ecofriendly statement. The composite materials developed in this study were not tested with respect to their degradation and ecotoxicity. Although the authors have replied to my former comments, that those studies will be the next research purpose, they are not addressed in the present manuscript, and for that I advise the authors to be careful when they state that the composites are degradable and ecofriendly. Even if some of the constituents of the composites are biodegradable and ecofriendly by their own, do not mean that the same will be applied to the final system. Moreover AZ91 alloy possesses a high amount of aluminum, an element that raises concerns about its safety for human health, and the silane used as a surface modifier, even in a small amount could induce toxicity.
Answer: Comparing with widely used copper alloy, aluminium alloy and ferroalloys containing variety of heavy metal elements, the magnesium alloys contains 90.31% Mg, 8.77% Al, 0.74% Zn, and 0.18% Mn. In corrosive medium, the magnesium alloys release less heavy elements. About a high amount of aluminum in magnesium alloys, there are many researches in biological implant materials in vivo and the magnesium Then the composite materials are more ecofriendly than the mostly used present mental alloys in industry applications. The composite materials contains less heavy elements and the basic component (PCL, ZnO and magnesium alloys) is with less toxicity. About the toxicity, the AZ magnesium alloys has done many researches in vivo and cell adhesion tests. (Thin Solid Films. 2010,518, 7563-7567& Materials Science & Engineering C. 2018, 90, 365-378). The results showed that the magnesium alloys has no significantly toxicity in vivo. The PFDTES is stable in the films of the composite materials and almost cannot release before the films damaged. The illustration of the Mg/P/Z/F/H materials’ degradation and biocompatibility has been added at the end of 3.3.
Regarding results section, the EIS measurements need to be discussed in more detail (time constants, physical meaning of those…). Since the authors decided to include “anticorrosion” in the manuscript title, the results section regarding this anticorrosion behavior should be deeper explored and explained. Moreover, the Nyquist graphs presented are not orthonormal, the time of immersion for which results are presented is not mentioned, the Rct values do not present units of measure.
Answer: Firstly, our coatings are organic and inorganic composite and the organic molecules covered the surface of the magnesium alloys’ surface. And the time constant of EIS is not the proper method to reflect the actual situation. So in our study, we use the EIS measurements in corrosive medium for appropriate stabilization time (about 40 min to 60min, has added in the manuscript) to test the composite materials. Furthermore, the corrosion process for a long time of the composite materials is complex with the slow degradation of composite materials and the Nyquist graphs for a long time of immersion in 3.5%wt NaCl are unable to present the real corrosion situation. We use EIS measurements to test the electrochemical behavior in corrosive medium at early protection process of the superhydrophobic films. The units of Rct including the units in Fig 5 have been modified and thank you very much for your kind comments.
About the polarization curves: can we assess the corrosion rate of Mg alloys by means of Tafel fitting of potentiodynamic polarization data? Is this logical at all? Please consult Progress in Materials Science 89 (2017) 92-193, section 2.1.1. Also the detailed information for Tafel fitting should be listed, such as the selected cathode region and computational method.
Answer: In Progress in Materials Science 89 (2017) 92-193, section 2.1.1., the authors clearly point out the polarization curves are useful for assessing approximate corrosion rates and the influence of various parameters such as alloying content. Furthermore, the icorr values inferred from potentiodynamic polarization curves for pure Mg/Mg alloys may not reflect the actual rate of
corrosion of the metal, since other parasitic chemical/electrochemical reactions may simultaneously occur at the Mg surface. And this is not a complete theory and more researches should be done about it in this field. For Tafel fitting is used the auto fitting with the CS studio 5 software. Firstly, the software automatically searches the region between about ±200mV of the open circuit potential. Then through the i=, use the calculation formula system to fit Ba and Bc and gain two calculation equation. After the software’s calculation, the calculation results are obtained including the i0, E0 and the corrosion rate.
In table 2 the units of corrosion rate are not correct, should be (mm/y) not (mm/a).
Answer: About the units of corrosion rate, there are several ways to write such as mm/year, mm/y, mm/Y, mm/yr, mm/a (Metals and Materials International. 22, 797-809 (2016)) and so on. The mostly used is mm/Y, mm/y and mm/yr. And in our revised the units of corrosion rate has been modified to mm/y in accordance with the comments.
Also in this section (3.3 Electrochemical corrosion behavior) the authors reach several conclusions, can you explain in detail how, from the polarization curves, it is possible to conclude that “(ii) the addition of PFDTES could improve the uniformity of the complex film and reduce porosity;” and “ (iii) the heating progress of Mg/P/Z/F/H materials could repair the gully defects and achieve molecular rearrangement. “
Answer: We revised our explainations about the change of the polarization curves’ part (3.3 Electrochemical corrosion behavior).
In the conclusions section is stated “In this study, completely degradable, ecofriendly and superhydrophobic Mg/P/Z/F/H composite materials were prepared successfully.” The results present can prove the preparation of the composites as well as their super hydrophobicity, but none of them gives any information regarding degradation and/or ecotoxicity. Correct the sentence or include results proving what is stated.
Answer: We modified our description in the revised manuscript and added the illustration of the Mg/P/Z/F/H materials’ degradation and biocompatibility in 3.3. The PCL, ZnO and magnesium alloys have the degradation functions and there is no new compounds produced in the composite materials. The toxicity about the magnesium, PCL and ZnO has done many works of their degradation and biocompatibility by the researchers. (Thin Solid Films. 2010,518, 7563-7567& Materials Science & Engineering C. 2018, 90, 365-378&Applied Surface Science.2017, 396, 249-258 &Materials Chemistry and Physics. 2021, 263, 124378 & Colloids and Surfaces B: Biointerfaces. 2021, 204, 111825) The composite materials are with less heavy metals and the main body of the materials has less toxicity. The surface modifier, PFDTES is stable in the composite materials with the high corrosion resistance and almost cannot release before the films destroyed. Thank you very much for your kind comments and it is essential and helpful for us to understand and improve our studies.
Reviewer 4 Report
Authors has failed to properly revise manuscript in accordance with reviewer’s comments. In particular:
Comments 1, 2, 3, 5c, 7 made for the first version of manuscript are still fully relevant for revised version.
Besides,
the reply to 5a still does not allow to understand how coating on metallic surfaces were studied by FTIR (whether total internal reflection was used or coating has been removed from surface etc).
Grammar still need to be improved.
Author Response
Authors has failed to properly revise manuscript in accordance with reviewer’s comments. In particular:
Comments 1, 2, 3, 5c, 7 made for the first version of manuscript are still fully relevant for revised version.
Comment 1:
Answer:
The composite materials contains less heavy elements and the basic component (PCL, ZnO and magnesium alloys) is with less toxicity and degradation. (Thin Solid Films. 2010,518, 7563-7567& Materials Science & Engineering C. 2018, 90, 365-378&Applied Surface Science.2017, 396, 249-258 &Materials Chemistry and Physics. 2021, 263, 124378 & Colloids and Surfaces B: Biointerfaces. 2021, 204, 111825) The AZ magnesium alloys has done many researches in vivo and cell adhesion tests. (Thin Solid Films. 2010,518, 7563-7567& Materials Science & Engineering C. 2018, 90, 365-378). The results showed that the magnesium alloys has no significantly toxicity in vivo. Further, the PFDTES is stable in the composite materials with the high corrosion resistance and almost cannot release before the films destroyed. The illustration of the Mg/P/Z/F/H materials’ degradation and biocompatibility has been added at the end of 3.3.
Comment 2:
Answer: Firstly, in this study we use the traditional and simple method, dip-coating, to prepare the composite coatings. If we introduced more literature review about these promising approaches such as PEO, laser texturing and so on, it is less helpful for readers to understand the simple way to prepare the materials in our points.
Comment 3:
Answer: The PFDTES, as the surface modifier, is stable in the composite materials with the high corrosion resistance and almost cannot release before the films destroyed. And the main body of the composite materials, PCL, ZnO and magnesium alloys are degradable and biocompatible. (Thin Solid Films. 2010,518, 7563-7567& Materials Science & Engineering C. 2018, 90, 365-378 & Materials Science & Engineering C. 2018, 90, 365-378 & Applied Surface Science.2017, 396, 249-258 & Materials Chemistry and Physics. 2021, 263, 124378 & Colloids and Surfaces B: Biointerfaces. 2021, 204, 111825).
Comment 5c:
Answer: In figure 3, the peak at 1172 cm–1 and 832 cm−1 of the Mg/P/Z/F/H line were detected in our FTIR. And we modified the FTIR statement of the peaks.
Comment 7:
Answer: The variation ranges of CA in wet atmosphere at different times are not over +1.8 and -1.6 degrees and we added the variation ranges of CA in the manuscript. And in our thoughts, the average level of the CA is useful and important to be referenced.
the reply to 5a still does not allow to understand how coating on metallic surfaces were studied by FTIR (whether total internal reflection was used or coating has been removed from surface etc).
Answer: The FTIR was tested by transmission spectroscopy using films removed from the surface of composite materials. In the FTIR test, the removed films and potassium bromide were used to form the pellet onto the assembled mold. The wafer was dried with an infrared lamp and put into the mold to measure by the transmission spectroscopy.
Grammar still need to be improved.
Answer: Thank you very much for you kind suggestions and comments. It is essential and helpful for us to modify the manuscript and improve our studies. Our revised manuscript has been added. If you have any problems please point out in your convenient time.
